# Frequency of Knee Pain and Risk Factors and Its Impact on Functional Impairment: A Cross-Sectional Study from Saudi Arabia

**DOI:** 10.3390/sports11090166

**Published:** 2023-09-01

**Authors:** Ali H. Alyami, Hussam Darraj, Khalid M. Hakami, Faisal Hakami, Mohammed Awaf, Nawaf Bakri, Sulaiman Hamdi, Abdulaziz Saber, Almuhanad Alyami, Mohammed Khashab, Abdulaziz H. Alhazmi

**Affiliations:** 1Department of Surgery, Ministry of the National Guard—Health Affairs, Jeddah 11426, Saudi Arabia; 2King Abdullah International Medical Research Center, Jeddah 22384, Saudi Arabia; 3Department of Surgery, King Saud bin Abdulaziz University for Health Sciences, Jeddah 22384, Saudi Arabia; 4Faculty of Medicine, Jazan University, Jazan 45142, Saudi Arabiaabalhazmi@jazanu.edu.sa (A.H.A.); 5College of Medicine, King Saud bin Abdulaziz University for Health Sciences, Jeddah 22384, Saudi Arabia

**Keywords:** knee pain, frequency of knee pain, exercise training, physical activity

## Abstract

Background: Adolescents frequently self-report pain, according to epidemiological research. The knee is one of the sites wherein pain is most commonly reported. Musculoskeletal disorders play a significant role in the prolonged disability experienced by individuals, leading to substantial global personal, societal, and economic burdens. Patellofemoral pain (PFP) is a clinical knee pain commonly affecting adolescents. This study aimed to estimate the frequency of knee pain in Saudi adolescents. Methods: This cross-sectional survey was conducted from June to November 2022 and included 676 adolescents aged 10 to 18 years. The participants were questioned regarding their demographics, school habits, and the impact of these factors on back pain, musculoskeletal pain in the past 12 months, as well as quality-of-life scale and knee pain symptoms. The data were analyzed using descriptive statistics, with frequencies and percentages presented for categorical variables. Analysis of variance (ANOVA) was performed to compare means between groups, while the chi-squared test was used to compare categorical variables. Statistical significance was set at *p* < 0.05. Results: A total of 676 adolescents participated in the study, with 57.5% females and 42.5% males. Among the participants, 68.8% were aged between 15 and 18 years. The prevalence of knee pain was notably higher among females (26%) compared to males (19.2%). Age and BMI were identified as significant predictors of knee pain. A significant association was also found between BMI classification and knee stiffness (*p*-value = 0.008). Furthermore, a significant difference was observed between adolescents who engaged in physical activities during leisure time and those who experienced difficulty bending (*p*-value = 0.03). Conclusions: Our study highlights a high prevalence of knee pain among Saudi adolescents, emphasizing the need for increased awareness about its risk factors. Preventive measures, including conservative approaches and lifestyle/activity modifications, can effectively mitigate adolescent knee pain.

## 1. Introduction

Musculoskeletal disorders contribute to disability and impose significant personal, societal, and economic burdens worldwide, and adolescence pain is commonly reported in epidemiological research, with the knee being a frequently affected site [1,2]. Knee pain is a prevalent clinical complaint among adults, with approximately half of the population over 50 years of age experiencing it [3].

The financial impact on the healthcare system is substantial, as one in six individuals with knee pain seeks medical attention annually, and a third of them become disabled [4]. Although less common in adolescents, knee pain in this age group is concerning as it may indicate the potential for more severe pain in the future [4].

The causes of knee pain in adolescents can vary from sudden, traumatic injuries to a gradual onset that may go unnoticed by both the adolescent and their parents [5]. Patellofemoral pain (PFP), a clinical knee condition, commonly affects physically active adults and adolescents. Patellofemoral pain syndrome (PFPS) is a leading cause of anterior knee pain in adolescents and young adults, characterized by peripatellar and retropatellar pain exacerbated by specific postures or activities [6].

PFP and ongoing knee pain can significantly limit adolescents’ physical activities. Consequently, reduced or discontinued physical leisure time activity due to knee pain may trigger a detrimental chain reaction, leading to decreased cardiorespiratory fitness, increased obesity, and adverse health outcomes [7,8]. Furthermore, adolescents who discontinue their involvement in sports during their teenage years may face an increased risk of developing diseases related to the heart and metabolism, as well as adopting sedentary lifestyles in adulthood [7,8].

Given the limited published reports on the epidemiology and associated factors of knee pain in Saudi adolescents, this study aims to determine the prevalence of knee pain among Saudi adolescents and investigate the factors related to its occurrence.

## 2. Subjects, Material, and Methods

### 2.1. Study Design, Location, and Setting

This cross-sectional observational study was conducted in Saudi Arabia, a country in the southwest of the Asian continent. With a population of 35 million, including 11 million adolescents, Saudi Arabia is bordered by the Red Sea to the west and the Arabian Gulf, United Arab Emirates, and Qatar to the east. Kuwait, Iraq, and Jordan border it to the north, while Yemen and the Sultanate of Oman border it to the south. This study adhered to the STROBE (Strengthening the Reporting of Observational Studies in Epidemiology) guidelines for reporting cross-sectional studies.

### 2.2. Participants

Participants were selected using a convenient sampling method, whereby individuals were recruited based on their accessibility and willingness to participate in the study. The sampling process involved reaching out to Saudi adolescents aged 10 to 18 years who resided in various regions of Saudi Arabia. The online questionnaire was administered via the Google Form platform between 30 October 2022 and 13 January 2022. Only investigators had access to the participants’ responses to ensure data quality and that frequent verification of the responses was conducted. The inclusion criteria for the study were Saudi adolescents aged 10 to 18 years residing in Saudi Arabia. Exclusion criteria included patients with known psychiatric illnesses, those who refused to participate, and non-Saudi adolescents living in Saudi Arabia. These criteria were established to ensure the study sample was representative of the target population and minimize the confounding factors’ impact.

### 2.3. Questionnaire

A self-administered online questionnaire was developed based on previously published and pre-tested questionnaires. The questionnaire consisted of three parts. The first part collected socio-demographic characteristics of participants, such as age, gender, height, and weight. The second part assessed factors related to school, including how participants carried their backpacks, the weight of their backpacks, physical education activities at school, and leisure-time physical activities. The third part of the questionnaire included an original tool to assess musculoskeletal pain over the previous 12 months and the Brazilian version of the Knee Injury and Osteoarthritis Outcome Score (KOOS) questionnaire to evaluate knee functional impairment. The third part also included the quality-of-life scale and a measure of knee pain symptoms.

### 2.4. Sample Size

The sample size for the study was calculated using a statistical formula for a cross-sectional survey design. The anticipated population proportion (p) was estimated to be 50% since no previous studies had been conducted in the region on knee pain among adolescents. Parameters for sample size calculation included a 95% confidence interval (CI), a marginal error of 5%, and a nonresponse rate of 20%. Based on these parameters, the final sample size was 385 adolescents. However, 676 participants were recruited for the study to increase statistical power.

### 2.5. Statistical Analyses

Data management and analysis were performed using SPSS version 23 (IBM Corp., Armonk, NY, USA). Continuous variables were analyzed using mean ± SD, and categorical variables were analyzed using frequency and percentages. Analysis of variance (ANOVA) was used to compare means between groups, while the chi-squared test was used to compare categorical variables.

### 2.6. Ethical Considerations

The study protocol was approved by the internal review board of the Ethics Committee of King Abdullah International Medical Research Center (Approval No.: NRJ22J/209/08, Date: 30 October 2022). Informed consent to participate was obtained before filling out the questionnaire via hospital records, and some information was obtained from the children’s parents. Each participant was provided with a serial number. Participation in the study was voluntary, and participants could withdraw if they wished.

## 3. Results

### 3.1. Socio-Demographic Characteristics of the Participants

A total of 676 adolescents participated in the study excluding 0.7% (n = 5) of participants who refused to participate. About 57% were girls and most participants (68.8%) fell into the age range of 15 to 18 years, and 26% were from the northern region. Regarding BMI, we followed the World Health Organization (WHO) guidelines for children and adolescents. Generally, a BMI below 18.5 is considered underweight, 18.5–24.9 is classified as normal weight, 25–29.9 is overweight, and 30 or above is classified as obesity [9]. Thus, 66% of participants had normal BMI. Half of the participants (49.9%) took physical education at school, while 11.7% engaged in physical activities during leisure time. Regarding backpack usage, 29.3% carried more than 10% of their body weight, and 74.4% had backpacks on their backs (Table 1).

### 3.2. Association of Knee Pain with Activities

Participants reported different activities associated with knee pain. The most commonly reported situation linked to knee pain was walking (21.9%), followed by climbing and descending stairs (20.6%). On the other hand, lying down was the least commonly reported situation associated with knee pain (4.9%) (Figure 1).

### 3.3. Prevalence and Factors Associated with Knee Pain

The frequency of knee pain among adolescents was found to be 45.3% (n = 306), with no significant difference between boys and girls. However, pain prevalence increased with age, as participants aged 12–14 years and 15–18 years showed a 30% and 55% higher likelihood of experiencing knee pain, respectively, compared to those aged 10 or 11 years (Table 2). Participants with a normal BMI reported less pain (25.1% vs. 40.5%; 95% CI: 1.147–3.298; *p*-value = 0.014) than those with abnormal BMI. Similarly, participants who carried ≤ 10% of their body weight in their backpacks reported less pain (42%; 95% CI: 1.364–2.665; *p*-value < 0.001). No significant differences were found in the other variable categories (Table 2).

### 3.4. Adjusted Analysis

After adjusting for confounding factors, age, BMI, and backpack weight remained associated with knee pain. Adolescents aged 15–18 years were 52% more likely to have knee pain compared to those under 12 years old (95% CI: 0.212–0.903; *p* = 0.025). Additionally, participants classified as underweight, overweight, or obese had a higher likelihood of experiencing knee pain compared to those with a normal BMI (95% CI: 1.032–2.999; *p*-value = 0.038). Those who carried more than 10% of their body weight in their backpacks were also more likely to experience knee pain than those who carried ≤ 10% (95% CI: 1.494–3.039; *p*-value < 0.001). The remaining variables did not significantly affect the outcome (Table 2).

### 3.5. Gender and Backpack Usage

There was a significant difference in knee pain between males and females, with females reporting significantly more pain than males (*p*-value = 0.003). Furthermore, there was a significant difference in pain scores based on how participants carried their backpacks. Those who carried their backpack on their back reported higher pain scores than those who carried them in their arms or hands (*p*-value = 0.02) (Table 3).

### 3.6. Association of Knee Stiffness and Everyday Activity

A significant difference in knee stiffness was observed according to age (*p*-value = 0.013) and BMI classification (*p*-value = 0.008). Differences were also found between BMI classification and everyday activity (*p*-value = 0.049), as well as between adolescents engaging in physical activities during leisure time and difficulty bending (*p*-value = 0.03). Additionally, a significant difference was found between physical activities during leisure time and knee stiffness (*p*-value = 0.01). There was a significant association between everyday activity and backpack weight (*p*-value = 0.004) and between backpack weight and quality of life (*p*-value = 0.019). Handling the backpack was also associated with difficulty in bending (*p*-value = 0.03) (Table 4).

## 4. Discussion

Epidemiological studies have consistently shown that adolescents frequently experience self-reported pain, with knee pain being one of the most prevalent areas of discomfort. Musculoskeletal problems, including knee pain, substantially burden individuals, society, and finances worldwide [10]. Therefore, it is crucial to understand adolescent knee pain and its causes and risk factors. The present study aimed to investigate the prevalence of knee pain among Saudi adolescents and its association with risk factors and functional impairment.

Our study revealed a prevalence rate of knee pain in adolescents over the past 12 months of 45.3%, consistent with a study conducted by Bhakti in India in 2021, where knee pain in adolescents was reported as 43.3% [11]. However, studies from Brazil in 2015 reported a lower percentage of knee pain among adolescents at 22.6% [10], while a study conducted in Finland in 1995 found a prevalence of 18.5% [12]. In England, a study in 1984 reported a frequency of self-reported knee pain among adolescents at 30% [13], which aligns with similar studies conducted in Denmark in September 2011, which found a knee pain rate of 27% [14,15]. Another Danish study reported a knee pain rate of 28% among adolescents [16]. In contrast, a study conducted in Canada reported a lower prevalence rate of knee pain among adolescents, at 7.4% [17]. The discrepancies in these percentages may be attributed to race, sample characteristics, and differences in awareness and reporting of knee pain symptoms. Further studies are warranted to explore this subject in more detail.

In the multivariate analysis of our study, we found no correlation between knee pain and gender. Similar findings were reported in a study conducted in Brazil [10], which revealed no difference in the prevalence of knee pain based on gender. However, girls demonstrated significantly more functional impairment than boys. Consistent results were also found in a study conducted in Canada [17]. The measurement of functional impairment may explain the discrepancy between the presence of an association and its absence. High-frequency symptoms, such as edema, fissures, difficulty bending, straightening stiffness, and limitations in daily activities, sports, and leisure activities, may account for this association. Spahn et al. also measured these symptoms in a study involving 2368 German youths [18].

Our findings indicate that knee pain increases with age among the evaluated teenagers, supported by previous studies conducted in Brazil and Finland [10,12]. Comparing musculoskeletal pain prevalence between children and adolescents, several studies have reported a significantly higher prevalence of knee pain among overweight individuals [17,18,19,20]. Our study aligns with these findings, as we found a correlation between knee pain and obesity in adolescents. However, a different study reported no impact of body weight on knee pain [10]. The more years of age, the more joints are exposed to mechanical stress which will evidently lead to more experience with pain. Again, this association is present in our current study. The high BMI is associated with more mechanical stress on lower limbs’ joints due to weight bearing load. This illustrates why it is associated with more pain, as shown in our study.

Participation in physical education at school has been shown to help prevent the onset of knee pain. Our study, although not finding a significant difference in this regard, revealed that adolescents with knee discomfort who did not participate in physical education exhibited worse functional impairment in the knee joint. Persistent musculoskeletal pain and physical inactivity have been associated in some studies [21]. In our study, no relationship was found between knee pain and physical activity at school, consistent with the Brazilian study’s findings [10]. Further studies employing various designs are needed to determine whether pain is a cause or a result of inactivity, as there is no consensus on the relationship between engaging in physical activity and the presence of knee pain.

Our study did not find an association between knee pain and engaging in physical activities during leisure time, which is consistent with Brazilian research [10]. Contrary to our expectations, there was no relationship between the extra weight of a backpack and knee pain in our study. However, we found an association between how the backpack is carried and knee pain. These results contradict the findings of another study [10]. Although the mean values of these factors did not show significant differences, there was a trend of greater functional impairment among adolescents with knee pain who carried heavier backpacks or carried them with their arms or hands. The literature on the additional weight of backpacks as a risk factor for knee joint pain presents conflicting findings. Some studies have shown that carrying heavy loads affects lower extremity joints, alters walking kinematics, increases quadriceps muscle fatigue, intensifies contact fatigue with the ground, and impairs the body’s ability to absorb stress during walking [22,23,24].

We found a relationship between the severity of knee pain and gender in adolescents, with women reporting higher pain intensity than men, which is consistent with findings from other studies [10,24]. Additionally, we observed an association between pain intensity and backpack-carrying style in adolescents, which has also been reported elsewhere [24]. Furthermore, our study revealed an impact of BMI on adolescents’ quality of life and physical activities, consistent with several previous studies [24,25,26,27]. However, one study reported contradictory results regarding this relationship [10]. We also found an association between knee pain symptoms, weight-bearing, and quality of life, indicating that carrying backpacks can cause discomfort among adolescents and impact their growth and development [24].

## 5. Limitations

Despite being one of the few studies in Saudi Arabia reporting the prevalence of knee pain, our study has some limitations. Being a cross-sectional study, there may have been limited time to cover all potential causes and other influencing factors. Future research should investigate additional factors associated with knee pain. Moreover, the study relied on a self-reporting questionnaire through a subjective data collection method (self-questionnaire), which may have been influenced by the respondents’ psychological and emotional state, potentially affecting the study outcomes. The validity and reliability of the study could be affected as we used a non-validated questionnaire. Moreover, it should be noted that we did not specifically investigate the relationship between patellar mal-tracking and knee pain, as well as its association with age and BMI. Future research endeavors should focus on exploring these factors to provide a more comprehensive understanding of their impact on adolescent knee pain. Lastly, the generalizability of our study’s results may be limited as the population was purposefully selected from a single location and on a small scale. Therefore, broader national research should be conducted to evaluate other factors associated with knee pain, and educational programs targeting reducing knee pain risk factors among adolescents could be developed.

## 6. Conclusions

In conclusion, our study revealed a high prevalence of knee pain among adolescents, with age and BMI playing significant roles. Older adolescents, mainly those aged 15 to 18 years, reported higher knee pain levels than younger age groups. Additionally, adolescents with a normal BMI experienced less knee pain than those who were overweight. There was a notable gender difference, with females reporting significantly more knee pain than males. The impact of backpack weight on knee pain was evident, affecting the quality of life of adolescents. Therefore, raising awareness about the risk factors associated with backpack weight and providing guidance on proper carrying techniques is crucial. Preventing adolescent knee pain can be achieved through conservative methods and minor lifestyle/activity modifications. Initial treatment often involves reducing or modifying activities that trigger knee pain. Physical therapy interventions can effectively manage knee pain, including quadriceps muscle stretching and exercises to improve strength, range of motion, and flexibility. Further research is needed to explore additional factors associated with knee pain among adolescents. Longitudinal studies and larger-scale investigations are warranted to comprehensively understand the causes, risk factors, and effective preventive measures for adolescent knee pain. By addressing these issues, we can promote adolescents’ well-being and functional ability, reducing the burden of knee pain in their daily lives.

## Figures and Tables

**Figure 1 sports-11-00166-f001:**
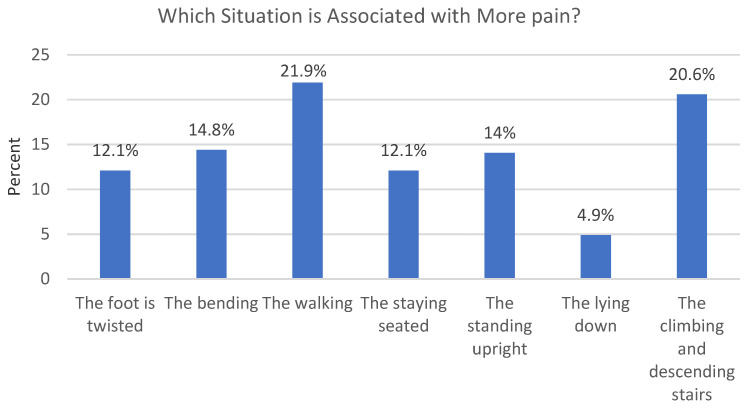
Presenting the most common situations that were reported to be associated with more knee pain.

**Table 1 sports-11-00166-t001:** Socio-demographic characteristics of the participants N = 676.

Variable	N	%
Gender		
Male	287	42.5%
Female	389	57.5%
Age		
10–11	43	6.4%
12–14	168	24.9%
15–18	465	68.8%
Place of residence		
The north region	179	26.5%
The eastern region	64	9.5%
The central region	141	20.9%
The western region	127	18.8%
The south region	165	24.4%
BMI		
Underweight	64	9.5%
Normal	444	65.7%
Overweight	118	17.5%
Obese	50	7.4%
Physical education at school		
Yes	337	49.9%
No	339	50.1%
Physical activity in leisure time		
Less than 6 h	218	32.2%
More than 6 h	79	11.7%
Insufficiently active	379	56.1%
Backpack weight		
>10% body weight	198	29.3%
≤10% body weight	478	70.7%
Form of carrying the backpack		
Back	503	74.4%
Arms/hands	173	25.4%

**Table 2 sports-11-00166-t002:** Binary logistic regression analysis for the frequency of knee pain with several predictor variables.

Knee Pain Frequency, Crude, and Adjusted Analysis of the Association between the Outcome and Selected Variables among Adolescents (n = 676)								
Predictor Variable	Knee pain &				Crude analysis OR (95% CI)	*p*-value *	Adjusted analysis OR (95% CI) #	*p*-value *
	Yes		No					
	n = 306	45.3%	n = 370	54.7%				
**Gender**								
Male (ref)	130	19.2	157	23.2	1		1	
Female	176	26	213	31.5	1.002 (0.738–1.3610)	0.989	1.146 (0.792–1.658)	0.468
**Age**								
10–11 (years) (ref)	13	1.9	30	4.4	1		1	
12–14 (years)	64	9.5	104	15.4	0.704 (0.342–1.449)	0.341	0.848 (0.398–1.809)	0.67
15–18 (years)	229	33.9	236	34.9	0.447 (0.227–0.878)	0.019 *	0.483 (0.212–0.903)	0.025 *
**BMI**								
Underweight (ref)	35	5.2	29	4.3	1		1	
Normal	170	25.1	274	40.5	1.945 (1.147–3.298)	0.014 *	1.759 (1.032–2.999)	0.038 *
Overweight	69	10.2	49	7.2	0.857 (0.464–1.583)	0.622	0.749 (0.400–1.402)	0.366
Obese	32	4.7	18	2.7	0.679 (0.318–1.450)	0.317	0.567 (0.261–1.232)	0.152
**Physical education at school**								
Yes (ref)	141	20.9	196	29	1		1	
No	165	24.4	174	25.7	0.759 (0.560–1.028)	0.075	0.785 (0.539–1.144)	0.208
**Physical activity in leisure time**								
Less than 6 h	91	13.5	127	18.8	1.185 (0.846–1.028)	0.324	1.143 (0.799–1.636)	0.465
More than 6 h	41	6.1	38	5.6	0.787 (0.484–1.278)	0.333	0.741 (0.442–1.241)	0.254
Insufficiently active (ref)	174	25.7	205	30.3	1		1	
**Backpack weight**								
>10% body weight (ref)	112	16.6	86	12.7	1		1	
≤10% body weight	194	28.7	284	42	1.906 (1.364–2.665)	<0.001 *	2.131 (1.494–3.039)	<0.001 *
**Form of carrying the backpack**								
Back (ref)	218	32.2	284	42.2	1		1	
Arms/hands	88	13	85	12.6	0.739 (0.523–1.045)	0.087	0.790 (0.539) 4–1.16	0.239

* *p*-value of 0.05 or less was considered to be a significant value; # adjusted odds ratio (AOR) was calculated through adjustment with all cofounders (demographic data and the rest of the predictor variables). & The predicted probability of membership was set by SPSS for no pain.

**Table 3 sports-11-00166-t003:** One-way ANOVA and t-independent tests were run to estimate the significant association between the means of a dependent continuous variable (pain score) and independent (demographic) variables.

Pain Score			
Variable	Mean	Standard Deviation	*p* Value *
Gender			
Male	42.38	20.75	0.003 *
Female	49.48	20.59	
Age			
10–11	41.53	16.25	0.068
12–14	41.71	23	
15–18	48	20.38	
BMI			
Underweight	49.71	22.29	0.297
Normal	47.17	20.21	
Overweight	42.46	19.88	
Obese	47.81	24.84	
Physical education at school			
Yes	45.53	21.36	0.46
No	47.28	20.58	
Physical activity in leisure time			
Less than 6 h	45.16	20.4	0.70
More than 6 h	46.82	22.07	
Insufficiently active	47.06	21.01	
Backpack weight			
>10% body weight	49.28	21.67	0.07
≤10% body weight	44.84	20.36	
Form of carrying the backpack			
Back	48.21	20.45	0.02 *
Arms/hands	42.15	21.57	

* The alpha criterion was considered significant at 0.05 or less.

**Table 4 sports-11-00166-t004:** Pearson chi-squared and Fisher’s exact tests were used for measuring the statistically significant associations of difficulty bending, stiffness of the knees, everyday activities, and quality of life with socio-demographic features.

Variable		Difficulty in Bending				Stiffness of the Knees				Everyday Activity				Quality of Life			
		Yes, More Affected n = 18, 100%	Yes, Less Affected n = 154	No n = 134	*p*-Value *	Yes, More Affected n = 17, 100%	Yes, Less Affected n = 113, 100%	No n = 176, 100%	*p*-Value *	Yes, More Affected n = 17, 100%	Yes, Less Affected n = 136, 100%	No n = 153, 100%	*p*-Value *	Yes, More Affected n = 30, 100%	Yes, Less Affected n = 147, 100%	No n = 129, 100%	*p*-Value *
Gender	Male	9 (50%)	67 (43.5%)	54 (40.3%)	0.68	9 (52.9%)	52 (46%)	69 (39.2%)	0.34	10 (58.8%)	56 (41.2%)	64 (41.8%)	0.37	12 (40%)	64 (43.5%)	54 (41.9%)	0.9
	Female	9 (50%)	87 (56.5%)	80 (59.7%)		8 (47.1%)	61 (54%)	107 (60.8%)		7 (41.2%)	80 (58.8%)	89 (58.2%)		18 (60%)	83 (56.5%)	75 (58.1%)	
Age	10 or 11 (years)	0 (0.0%)	8 (5.2%)	5 (37%)	0.78	3 (17.6%)	3 (2.7%)	7 (4%)	0.013 *	0	9 (6.6%)	4 (2.6%)	0.33	2 (6.7%)	8 (5.4%)	3 (2.3%)	0.42
	12–14 (years)	3 (16.7%)	34 (22.1%)	27 (20.1%)		7 (41.2%)	23 (20.4%)	34 (19.3%)		3 (17.6%)	31 (22.8%)	30 (19.6%)		5 (10.7%)	35 (35%)	24 (18.6%)	
	15–18 (years)	15 (83.3%)	112 (73.7%)	102 (76.1%)		7 (41.2%)	87 (77%)	135 (76.7%)		14 (82.4%)	96 (70.6%)	119 (77.8)		23 (76.7%)	104 (70.7%)	102 (79.1%)	
BMI	Underweight	2 (11.1%)	17 (11%)	16 (11.9)	0.2	0 (0%)	8 (7.1%)	27 (15.3%)	0.008 *	1 (5.9%)	16 (11.8%)	18 (11.8%)	0.049 *	4 (13.3%)	17 (11.6%)	14 (10.9%)	0.03 *
	Normal	8 (44.4%)	79 (51.3%)	83 (61.9%)		6 (35.3%)	62 (54.9%)	102 (58%)		7 (41.2%)	66 (48.5%)	97 (63.4%)		13 (43.3%)	70 (47.6%)	87 (67.4%)	
	Overweight	4 (22.2%)	38 (24.7%)	27 (20.1%)		6 (35.3%)	31 (27.4%)	32 (18.2%)		5 (29.4%)	37 (27.2%)	27 (17.6%)		6 (20%)	45 (30.6%%)	18 (14%)	
	Obese	4 (22.2%)	20 (13%)	8 (6%)		5 (29.4%)	12 (10.6%)	15 (8.5)		4 (23.5%)	17 (12.5%)	11 (7.2%)		7 (23.3%)	15 (10.2%)	10 (7.8%)	
Physical education	Yes	10 (55.6%)	68 (44.2%)	63 (47%)	0.62	9 (52.9%)	52 (46%)	80 (45.5%)	0.839	9 (52.9%)	68 (50%)	64 (41.8%)	0.321	13 (43.3%)	73 (49.7%)	55 (42.6%)	0.48
	No	8 (44.4%)	86 (55.8%)	71 (53%)		8 (47.1%)	61 (54%)	96 (54.5%)		8 (47.1%)	68 (50%)	89 (58.2%)		17 (56.7%)	74 (50.3%)	74 (57.4%)	
Physical activity in leisure time	Less than 6 h	4 (22.2%)	39 (25.3%)	48 (35.8%)	0.03 *	4 (23.5%)	29 (25.7%)	58 (33%)	0.01 *	4 (23.5%)	39 (28.7%)	48 (31.4%)	0.4	10 (33.3%)	39 (26.5%)	42 (32.6%)	0.68
	More than 6 h	6 (33.3%)	20 (13%)	15 (11.2%)		6 (35.3%)	20 (17.7%)	15 (8.5%)		3 (17.6%)	23 (10.9%)	15 (9.8%)		4 (13.3%)	23 (15.6%)	14 (10.9%)	
	Insufficient	8 (44%)	95 (61.7%	71 (53%)		7 (41.2%)	64 (56.6%)	103 (58.5%)		10 (58.8%)	74 (54.4%)	90 (58.8%)		16 (53.3%)	85 (57.8%)	73 (56.6%)	
Backpack weight	>10%	7 (38.9%)	65 (42.2%)	40 (29.9%)	0.09	8 (47.1%)	46 (40.7%)	58 (33%)	0.26	7 (41.2%)	63 (46.3%)	42 (27.5%)	0.004 *	11 (36.7%)	65 (44.2%)	36 (27.9%)	0.019 *
	≤10%	11 (61.1%)	89 (57.8%)	94 (70.9%)		9 (52.9%)	67 (59.3%)	118 (67%)		10 (58.8%)	73 (53.7%)	111 (72.5%)		19 (63.3%)	82 (55.8%8	93 (72.1%)	
Form of handling the backpack	Back	12 (66.7%)	123 (79.9%)	83 (61.9%)	0.03 *	10 (58.8%)	82 (72.6%)	126 (71.6%)	0.5	10 (58.8%)	106 (77.9%)	102 (66.7%)	0.054	20 (66.7%)	111 (75.5%)	87 (67.4%)	0.28
	Arm/Hands	6 (33.3%)	31 (20.1%)	51 (38.1%)		7 (41.2%)	31 (27.4%)	50 (28.4%)		7 (41.2%)	30 (22.1%)	51 (33.3%)		10 (33.3%)	36 (24.5%)	42 (32.6%)	

* The alpha criterion was considered significant at 0.05 or less.

## Data Availability

The data presented in this study are available upon request from the corresponding author.

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
