# Peer review of "Frequency of Knee Pain and Risk Factors and Its Impact on Functional Impairment: A Cross-Sectional Study from Saudi Arabia"

_sports, 2023, doi:10.3390/sports11090166_

Round 1
Reviewer 1 Report (New Reviewer)
The framework of manuscript is intact and English writing is fluent. This issue should be important and interesting, and reading is enjoying. However, the methodology used in this study is so rough that reliability and validity are doubtful. Using online investigation with questionnaire to study knee pain are generally questionable in precision. The wrong conclusion often occurs. Under equivocal grouping, this study utilizing statistical techniques to achieve significance is easily mis-leading.
Some doubts require clarification:
1. In Abstract, Background, L4-5: Insidious --- [5]. What do you mean? Generally, references should not appear in Abstract,
2. The qualified knee pain in this study must be clearly defined first. It should be the most critical key to decide whether or not this study is believable. Duration, localization, intensity, situation on attacking, and with or without associated knee swelling must be distinguished. Or, only with or without knee pain being roughly grouped for study will make wrong conclusion.
3. The causes of knee pain in adolescents include evident or insidious trauma, benign or malignant tumors, active or latent infection, metabolic or immunologic origins, or congenital or developmental anomalies. Especially, patellar mal-tracking is popular. The predisposing factors are so complex that grouping for comparison must be very reasonable.
4. In Discussion, the reasons for age and BMI causing knee pain should be explained reasonably. Theoretically, both are unrelated to knee pain.
5. Without evident trauma histories, patellar mal-tracking has gradually been believed to cause knee pain in the majority of adolescents in the literature. Patellar mal-tracking is unrelated to age and BMI. Please comment your findings in Discussion.
6. References are written in a mess. Full spelling or formal abbreviation of journals must be used. The styles must be consistent. e.g., page 122-129 or 122-9, capital or lowercase.
The whole are largely satisfactory, but some errors need re-checking.
Author Response
The authors thank the reviewer for this comment.

Reviewer 2 Report (New Reviewer)
Title: spelling is not correct and should be improved. In its present form makes no complete meaning.
Abstract: Background
2nd sentence: Musculoskeletal disorders contribute…this statement is not relevant.
Methods:
The first 2 sentences of the results belong to the methods section.
Describe in brief the parameters you explored.
Results:
Provide always standard deviations when you are reporting measurements or frequencies.
Introduction
The first 2 paragraphs should be merged. Start with the description of the burden of the musculoskeletal disorders in general and then focus in the adolescent group.
Paragraph 3:
The etiology of adolescent knee pain…..Needs to be restructured.
cardiometabolic diseases…..rephrase please
Given the limited published reports…..refer to the limited similar reports or other similar research papers.
Participants:
Please report in more detail the method you used to choose your sample.
Those refusing to participate are not part of the exclusion criteria. The percentage of dropouts should be reported.
The Brazilian version of KOOS was used. Is there any version validated in Arabic which would be more appropriate to use?
Statistical Analyses
Use a smaller font size.
3.3.
Report the incidence of knee pain in subjects with an abnormal BMI
Please define the limits of abnormality of BMI
Please define and describe how where you able to report on the parameter called: Stiffness of The Knees. Do you think a young adolescent is able to decide what stiffness is?
The discussion should be more compact and be limited on the results of the study.
The Conclusion section should be only 4-5 lines long.
Table 1. is there any anticipated difference in adolescent residing in the various geographic parts of the place of their residence?
mild language editing is indicated to make the paper more presentable, although it is already well-written.
Author Response
The authors thank the reviewer for this comment.

Round 2
Reviewer 1 Report (New Reviewer)
After revision, parts of comments have been replied and responded well. However, parts of comments are still not corrected. In the whole, this manuscript is satisfying.
1. Despite that authors do not respond for definition of knee pain used in this study, the content of manuscript may still be acceptable.
2. Why the age or BMI is the risk factor to cause knee pain? These findings should be explained in Discussion. It is the validity of this study. Without evident knee trauma, degenerative change normally occurs after 50 years. In this study, all subjects are adolescents with less than 18 years. Therefore, the authors’ explanation is doubted.
3. Numerous errors in References are still not corrected and need correction: Space between words (refs. 1,6,7,22,23), punctuation (refs. 3,4,5,8,9,13, ---), page style (ref. 10,15), capital or lowercase (ref. 11,17,24), journal abbreviation (ref. 9, ---), -- etc.

Author Response
Thanks for this comment.

This manuscript is a resubmission of an earlier submission. The following is a list of the peer review reports and author responses from that submission.
Round 1
Reviewer 1 Report
This is a cross-sectional study, with adolescents aged 10 to 18 years, with the objective of estimating the prevalence of knee pain in Saudi adolescents.
The manuscript is well written, however it does not seem to achieve the objective of estimating the prevalence of knee pain in Saudi adolescents.
Prevalence is a measure used to estimate the frequency of an event in the population. Therefore, this frequency measure requires a population sample.
It was not possible to assess whether the sample complies with the requirements to be considered a population, as the parameters to estimate the sample size were not described, nor evidence that the sample was constructed to represent the population.
Authors are requested to inform more details of the sample. If the population sample requirements have not been taken into account, it is suggested to replace the term prevalence with another suitable one (frequency, for example).
Author Response
I fix in nwe submiit
Reviewer 2 Report
First of all, thank you for the opportunity to review this very interesting article.
Here are some recommendations:
Abstract
-Review punctuation issues in the summary, such as after "Background" or "ages 10 to 18". Also the last two lines and keywords are in a different format.
Introduction
- "Sidious" should not begin with a capital letter.
Material and Methods
- Saudi Arabia cannot have a population of 3 million with 11 million teenagers, please correct this aspect.
- During the description of the questionnaire they have forgotten to explain the fourth section of the questionnaire.
- I advise you to review the writing and structure of the subsection "Data collection" after "In the following influencing factors...". As the wording becomes difficult for the reader to understand.
- Please include the complete and correct SPSS citation.
Results
- Please indicate how you categorize BMI in the Methods section.
- The tables are not in the format required for the journal.
- In the title of Table 1, the sample number must appear in parentheses.
- Figure 1 must be centered.
- Please simplify the second row of Table 2 and use the same format for all letters and numbers.
- Please reduce the size of Table 4 to make it easier to read and write the title before it.
Discussion
- Please use synonyms and write a story in the discussion. The second paragraph just repeats "According to" with numerous punctuation errors.
In summary:
1) Use the current journal format, the one delivered is 2020.
2) Check the wording and punctuation.
3) Check the tables
Author Response
I fix in new submit

Round 2
Reviewer 1 Report
This is the revised version of the manuscript "Prevalence of Knee Pain and Risk Factors and Its Impact on Functional Impairment among Saudi Adolescents".
Thanks to the authors for reviewing the notes of the previous report.
However, the changes made generated other suggestions:
- Section 2.1 "This study will be carried out in Saudi Arabia, which has a population of 3 million, with 11 million adolescents".
Review the values.
In section 2.2: "The incidence rate of each of the following influencing factors was measured (...) The impact of each of them on the demographic was measured".
As this is a cross-sectional study, how was the incidence rate calculated? Apparently, the described methodology is not sufficient to calculate the incidence rate. It is not clear what was the zero time (t0, baseline) for observation of the free outcome and the longitudinal follow-up to estimate the incidence rate. The type of study design described apparently makes it impossible to calculate the incidence rate.
It is also not clear how it was possible to measure impact in a cross-sectional study. This type of measurement is usually done in longitudinal studies.
Results Section:
- Standardize the number of decimals used in the text and tables.
Author Response
resubmit

Reviewer 2 Report
Introduction
there are still different assertions by the authors without any reference:
- "Given that one in six people with knee pain will have at least one medical appointment each year and that one-third of them will be disabled, knee pain has a significant financial impact on the health system"
- "Reduced or discontinued physical leisure time activity due to knee pain may set off a chain reaction in which decreased physical activity leads to poor cardiorespiratory fitness, increased obesity, and poor health consequences"
Subjects, Material and Methods
Please note my previous corrections. There cannot be 11 million teenagers in a population of 3 million inhabitants.
"This study will be carried out in Saudi Arabia, which has a population of 3 million, with 11 million adolescents"
Discussion
Again, please note my previous recommendations. In the second paragraph of the discussion uestdes repeat "According to" 3 times in total. Please rephrase it as follows.
"Our research found no links between a backpack’s extra weight and knee pain, but it uncovered links between how the rucksack is carried and the presence of knee pain. This is contrary to the results of another study." Please add references
Author Response
resubmit

Round 3
Reviewer 1 Report
Dear authors, thank you for the review.
As informed since the first report, it is not possible to estimate prevalence using a convenience sample. Prevalence is a measure of population frequency obtained by a probabilistic sample capable of representing this population. I suggest replacing the term "prevalence". My suggestion is to use "frequency" in all your results and title.